# Molecular and Genetic Mechanism of Non-Syndromic Congenital Cataracts. Mutation Screening in Spanish Families

**DOI:** 10.3390/genes12040580

**Published:** 2021-04-16

**Authors:** Celia Fernández-Alcalde, María Nieves-Moreno, Susana Noval, Jesús M. Peralta, Victoria E. F. Montaño, Ángela del Pozo, Fernando Santos-Simarro, Elena Vallespín

**Affiliations:** 1Department of Ophthalmology, Hospital Universitario La Paz, 28046 Madrid, Spain; marianievesmoreno@gmail.com (M.N.-M.); sunoval@gmail.com (S.N.); jesuspc22@hotmail.com (J.M.P.); 2Department of Molecular Ophthalmology, Medical and Molecular Genetics Institute (INGEMM) IdiPaz, CIBERER, Hospital Universitario La Paz, 28046 Madrid, Spain; victoriaeugeniafdezmontano@hotmail.com (V.E.F.M.); elena.vallespin@salud.madrid.org (E.V.); 3Department of Clinical Bioinformatics, Medical and Molecular Genetics Institute (INGEMM) IdiPaz, CIBERER, Hospital Universitario La Paz, 28046 Madrid, Spain; ingemm.adelpozo@gmail.com; 4Department of Clinical Genetics, Medical and Molecular Genetics Institute (INGEMM) IdiPaz, CIBERER, Hospital Universitario La Paz, 28046 Madrid, Spain; fernando.santos@salud.madrid.org

**Keywords:** congenital cataracts, genetics, next-generation sequencing, ophthalmogenetics

## Abstract

Our purpose was to identify mutations responsible for non-syndromic congenital cataracts through the implementation of next-generation sequencing (NGS) in our center. A sample of peripheral blood was obtained from probands and willing family members and genomic DNA was extracted from leukocytes. DNA was analyzed implementing a panel (OFTv2.1) including 39 known congenital cataracts disease genes. 62 probands from 51 families were recruited. Pathogenic or likely pathogenic variants were identified in 32 patients and 25 families; in 16 families (64%) these were de novo mutations. The mutation detection rate was 49%. Almost all reported mutations were autosomal dominant. Mutations in crystallin genes were found in 30% of the probands. Mutations in membrane proteins were detected in seven families (two in *GJA3* and five in *GJA8*). Mutations in *LIM2* and *MIP* were each found in three families. Other mutations detected affected *EPHA2, PAX6, HSF4* and *PITX3*. Variants classified as of unknown significance were found in 5 families (9.8%), affecting *CRYBB3, LIM2, EPHA2, ABCB6* and *TDRD7*. Mutations lead to different cataract phenotypes within the same family.

## 1. Introduction

The ocular lens is part of the anterior segment of the human eye. As part of the refractive system, its main function is to transmit and focus light onto the retina due to its transparency, shape and accommodative power [1,2,3]. Proteins represent up to 60% of the lens mass, mostly as lens crystallins, whose distribution and stability are critical for light transmission [4,5]. Any disturbance in the cell structure will produce diffraction, absorbance or reflection of the light, which will result in diminished vision [6].

A cataract is an opacity of the lens [7]. Congenital or infantile cataracts present within the first year of life and are considered to be the main cause of treatable blindness during infancy worldwide. Reported incidence ranges from 12 to 136 per 100,000 births, with higher rates in under-developed countries [8]. Congenital cataracts can be caused by connatal infections, fetal suffering or genetic variations. It is estimated that 8.3–25% of congenital cataracts are hereditary [9]; remarkably, they were the first autosomal disease mapped in humans [10]. Hereditary cataracts can be divided into two groups: syndromic cataracts, which form part of a systemic disease affecting other tissues, and non-syndromic cataracts, in cases in which the lens is the only compromised organ. It is important to understand that cataract phenotype is not a reliable indicator of the affected gene or mutation, since identical cataracts can be caused by mutations at different loci; the same mutation can also cause different types of cataracts [10,11,12,13,14,15].

Currently, the only treatment for congenital cataracts is surgery, with the removal of the opacified lens. Early intervention is crucial due to the high risk of amblyopia. Cataract surgery in babies and infants has a high risk of postoperative complications and it is a challenge to achieve a good visual acuity despite timely intervention and adequate follow-up [16,17,18,19,20].

Mutations causing congenital cataracts affect the process of protein aggregation and involve the breakdown of lens architecture. All three types of Mendelian inheritance patterns have been reported, although the most frequent is autosomal dominant with high penetrance. There may be remarkable phenotype variability within a family and even between the two lenses of the same individual (Figure 1a polar cataract; Figure 1b nuclear cataract). Mutations in over 52 genes have been described so far, 35 associated with non-syndromic congenital cataracts and 17 which may cause either syndromic or isolated cataracts. Mutations can be divided depending on the proteins affected into four main groups: lens crystallins (37%), membrane proteins (22%), transcription factors (14%) and cytoskeletal proteins (3–7%) [21,22,23,24,25].

The purpose of this study is to identify the mutations leading to the development of non-syndromic congenital cataracts in our hospital, which is one of the main referral centers for the diagnosis and treatment of this pathology in Spain, through the implementation of next-generation sequencing (NGS) for mutation screening.

## 2. Objectives

-To evaluate the implementation of NGS for mutation screening in congenital cataracts at a tertiary hospital in Spain.-To identify the mutations that produce congenital cataracts in Spain and compare them with those reported for other populations.

## 3. Results

A total of 62 probands from 51 families diagnosed with non-syndromic congenital cataracts were recruited. After performing NGS analysis with a specifically designed panel, we identified pathogenic or likely pathogenic variants in 32 patients and 25 families: this represents a mutation detection rate of 49% (Table 1 and Table 2). Variants classified as a variant of unknown significance (VUS) were found in seven patients and five families (9.8%).

Gender distribution was 35.5% of male and 64.5% of female probands. The majority of families were from a Caucasian European background. The most frequent cataract phenotype was nuclear (55.3%), followed by lamellar (23.7%), posterior subcapsular (5.3%) and posterior polar (2.6%). In 30% of subjects there was a further anatomical ocular malformation such as microphthalmia (20%), microcornea (10%) or aberrant iris anatomy (6.67%). A previous family history of congenital cataracts was reported in 44.7% of probands, with a likely autosomal-dominant mode of inheritance. None of the families reported consanguinity. One family (Family 27) showed a compound heterozygous inheritance pattern, the only case in our cohort.

Out of the 25 families with a likely pathogenic or pathogenic variant, 16 were de novo mutations (64%). Of all the identified variants, 30% were mutations in crystallin genes (one each in genes *CRYBB2*, *CRYBA4*, *CRYGS*, *CRYAA* and *CRYBB3* and two each in genes *CRYGD* and *CRYGC*). All these mutations have already been reported and therefore were considered as likely pathogenic or pathogenic, except for one case which was interpreted as a VUS variant. The most frequent phenotype was nuclear cataract, and four families had another associated anterior chamber malformation: two had microphthalmia and microcornea (Families 2 and 4) and two others iris hypoplasia (Families 7 and 8).

Mutations in membrane proteins were detected in seven families (two in *GJA3* and five in *GJA8*). Cataract phenotype was predominantly lamellar in *GJA3* and nuclear in *GJA8* mutations. Three of the seven families had microphthalmia and/or microcornea (Families 12, 15 and 16). All mutations were classified as likely pathogenic or pathogenic. The third most frequent mutation affected *LIM2* (10%), detected in three different families (Families 17, 18 and 19). Cataract phenotype was nuclear or lamellar, except for one proband with posterior polar cataract. None of them had any other ocular malformation.

Mutations in *MIP* were also found in three families (Families 24, 25 and 26). They all presented with nuclear cataracts with no other ocular malformations and all mutations were classified as likely pathogenic or pathogenic.

Two probands who had nuclear cataracts plus microphthalmia and microcornea had a mutation in *EPHA2*; one of them was considered as likely pathogenic (Family 20). Mutations in *PAX6* were found in two probands, one of them with isolated congenital cataracts (Family 22) and the other one with associated anterior chamber malformations (Family 23). Other pathogenic variants found in one proband each, were the following: *HSF4* (Family 27) and *PITX3* (Family 28). Other variants found among probands and classified as VUS (ACMG) were *ABCB6* (Family 29) and *TDRD7* (Family 30).

### 3.1. LIM2 Mutations in Two Spanish Families

Mutations in *LIM2* were identified in Family 17 (Figure 2a) and Family 18 (Figure 2b). *LIM2* belongs to a group of small integral membrane glycoproteins with a crucial role in lens biology, cytoskeletal integrity, cell morphology and intercellular communication [26]. The product of *LIM2* is a 173-amino-acid membrane protein named MIP20 (alternatively known as MP17/MP18/MP19), the second most abundant integral membrane protein of the ocular lens fiber cells of vertebrates [27].

Family 17 includes six affected individuals belonging to three different generations, as shown in Figure 2a, all of them Caucasian Europeans with no known consanguinity risk factors. The first generation affected members reported being diagnosed in early infancy; the second- and third-generation members had been diagnosed in our center during their first year of life. Nystagmus was present in four of the patients and amblyopia in three of them. Cataract phenotype was lamellar (four patients) and posterior polar (one patient) and one of them also suffered from glaucoma. No anterior chamber or posterior pole malformations were present. NGS was performed, and a *LIM2* (c.388C>T) heterozygous mutation was found in all affected members.

Family 18 includes five affected individuals belonging to three different generations (Figure 2b) all of them Caucasian Europeans. Affected patients of the second and third generations had been diagnosed with congenital cataracts during their first year of life. All patients had been operated promptly after diagnosis. None of them presented nystagmus or glaucoma. Cataract phenotype was nuclear in all cases and no anterior chamber or posterior pole malformations were present. NGS was performed, and a *LIM2* (c.388 C>T) heterozygous mutation was found in all affected members.

### 3.2. Compound Heterozygous HSF4 Mutation in a Spanish Family

A mutation in *HSF4* was identified in Family 27 (Figure 3). The heat shock factor (HSF) family of genes encode transcriptional regulators and mutations have been found to be inherited in both autosomal dominant and recessive patterns. In a study conducted by Merath et al. the evaluation of mutant *HSF4* proteins led to the conclusion that autosomal recessive mutations result in loss of regulatory domains present at the C-terminal end of the protein [28]. Family 27 included one affected individual, as shown in Figure 3. All members were Caucasian Europeans. The proband had been diagnosed with congenital cataracts during her first year of life. Cataract phenotype was nuclear, and no anterior chamber or posterior pole malformations were present. NGS was performed, and a *HSF4* (Allele 1: 486-2A>G, Allele 2: 1302delG) compound heterozygous mutation was found in the proband.

## 4. Discussion

In the present study, we performed a mutational analysis of Spanish patients diagnosed with congenital cataracts implementing the NGS technique. The main reason for choosing the NGS technique versus whole exome sequencing, was that it is less expensive while providing an efficient analysis with fewer incidental findings, therefore making it more suitable for a tertiary hospital. Besides, whole exome sequencing can be performed in cases in which the NGS technique does not identify mutations. The main advantage of custom targeted sequencing is the possibility of tailoring it to specific needs, by including a complete gene sequence, or specific intronic sequences or else UTR regions.

Mutation detection rate was 49% in the 51 families included in the study; this figure is slightly lower than reported for similar studies, which ranged from 58% to 70% [29,30,31,32]. The detection rate was higher than reported with Sanger sequencing [33]. There was a high prevalence of VUS variants in our cohort (9.8%). Almost all reported mutations were autosomal dominant, and approximately 64% of mutations detected were de novo, a lower rate than previously reported [34,35]. Only one compound heterozygous mutation was found in our cohort [36].

Crystallins α, β and γ are the predominant proteins of the lens and essential to its refractive properties [19]. Dominant inheritance has been reported for mutations in all 12 crystallin genes and recessive inheritance has also been reported for mutations in 5 of them. In crystallin genes, non sense-mediated decay is determinant to whether the mutant alleles will have recessive or dominant effects. Thus, truncating mutations which appear in the initial coding sequences of the gene will be affected by non sense-mediated decay and follow a recessive pattern; whereas mutations in the final exon will be associated with dominant disease [16]. Mutations in crystallin genes accounted for the majority of hereditary congenital cataracts in our study (30%), which was an expected result as this is the most frequent mutation reported up to date. Still, the percentage was lower than in other published articles. In two studies performed in Chinese families, mutations in the crystalline genes represented 67% [33] and 40% of the pathogenic mutations detected [37]. Hansen et al. in their study including twenty-eight Danish families found that mutations in genes encoding crystallins and connexins accounted for 53.5% of pathogenic variants, although only 17 congenital cataracts genes were screened [38]. Therefore, it seems that the genes affected by pathogenic mutations vary widely and may depend on the race of the subjects. Two families (Families 7 and 8) presented a variant in *CRYGC*. This mutation changes protein conformation, decreasing its solubility and stability, as well as affecting its interaction with other crystallins, increasing aggregation. Both families presented with nuclear congenital cataracts and iris malformations, with microphthalmia in one of them. Mutations in *CRYGC* have previously been reported in patients with nuclear or lamellar cataract, microcornea and microphthalmia [32].

The absence of *LIM2* is associated with accelerated breakdown of cytoskeletal proteins in central lens cells and cataractogenesis, as shown in *LIM2*-deficient mice models designed by Shi et al. and Steele et al. [26,28]. Missense mutations have been associated with autosomal recessive cataracts (c.313T>G (p.Phe105Val); c.587G>A (p.Gly154Glu); c.233G>A (p.Gly78Asp)) [27], and one missense heterozygous mutation causing membranous, lamellar and nuclear cataract (c.388C>T) has recently been reported in European and Asian populations [29,30]. In our cohort, we identified two families presenting with this variant. In accordance with previous reports, none of these families’ patients presented any anterior chamber malformation, and cataract phenotype was diverse (membranous, nuclear, lamellar, posterior polar). This leads us to suspect there might be a higher prevalence of this mutation than has previously been reported. Further studies investigating *LIM2* should be performed, and this gen should be considered when performing congenital cataracts genetics study.

Special attention should be devoted to *HSF4*, the only compound heterozygous variant found in the studied families. The most common cataract phenotype associated with this gene is lamellar, and no anterior chamber malformations have been reported so far [39]. Family 27 had risk factors for consanguinity, and presented with nuclear cataract with no anterior chamber malformations. In the European population, three different autosomal dominant mutations have been found so far, in Danish [38] and British families [40], showing predominantly a lamellar phenotype. Jiao et al. described a Pakistani consanguineous family with nuclear cataract and concluded that recessive mutations are mainly located in hydrophobic repeats (HR-A/B) or downstream of the hydrophobic repeat, in regions important for a trimeric formation and transcriptional activation of *HSF4* [41].

Connexins are crucial for maintaining cell-to-cell communication, which in the lens occurs mainly via gap junction channels made up of the isoforms *GJA1*, *GJA3* and *GJA8*. Currently, 55 heterozygous variants and 1 homozygous variant have been found in *GJA3* as well as 90 heterozygous variants and 1 homozygous variant in *GJA8* [16]. The majority of mutations in *GJA8* are dominant missense mutations, and sometimes lead to corneal abnormalities together with congenital cataracts [16]. Gap junction genes (*GJA8* and *GJA3*) represent the second most common group of genes affected, as in Shielsa et al. and Zhang et al. [9,33]. We found seven families presenting this variant, with nuclear and lamellar cataracts. Among them, three had anterior chamber malformations, which is a higher prevalence than in other series.

The variant identified in *EPHA2* has previously been reported in Mexican [32], Australian [42] and British [43] families. Experimental studies have shown that this variant creates an aberrant splicing signal originating a mutant protein with additional amino acid residues. Family 20 had nuclear cataracts associated with microphthalmia, as opposed to the families included in the afore-mentioned reports, which had no anterior chamber malformation. However, intrafamilial variable expression has been reported and therefore a complex genotype–phenotype correlation is presumed [32].

The pituitary homeobox (*PITX*) family of proteins are transcription factors that contain two domains. Mutations in the *PITX* family gene *PITX3* cause congenital cataract and anterior segment mesenchymal dysgenesis, such as corneal opacity, microphthalmia and microcornea [39]. In our cohort, we were able to identify one heterozygous variant in *PITX3* which had previously been described as a homozygous mutation [44]. Family 28 presented with a novel mutation leading to posterior subcapsular cataract without any other ocular dysgenesis; this phenotype had already been corelated with *PITX3* in English and Chinese families [45,46]. On the other hand, posterior subcapsular cataract due to *PITX3* mutation can also coexist with anterior chamber malformation as shown in five Belgian families [47]. 

It should be taken into account that the same mutations may lead to different phenotypes within the same family. We believe that all affected subjects with a known mutation should be evaluated, in order to further characterize variable expression by a same mutation in a given gene.

One limitation of our study is that the NGS panel we used would need to be updated with the new genes for which previously unknown mutations are described. Another limitation is the relatively low number of patients included, as well as the lack of mutation functional analysis. This might explain why our mutation rate is slightly under the mean value reported so far.

Cases without a molecular diagnosis can be explained because the causative gene might not be included in the panel design. There may be genes encoding proteins involved in congenital cataracts of a specific biochemical marker that is currently unknown or not related to human disease.

## 5. Materials and Methods

The present study was approved by the Ethics Committee of La Paz University Hospital of Madrid, and followed the tenets of the Declaration of Helsinki. Informed consent was obtained from all patients, parents or legal tutors. Our center’s electronic medical records were used to identify all patients diagnosed with congenital cataracts between 2005 and 2020. Our center is a major tertiary referral hospital in Spain and patients with suspected congenital cataracts are referred to us for diagnosis confirmation and treatment from the whole central region of the country. For genetic diagnosis, patients have been included in the diagnostic algorithm developed and shown in Figure 4, although it should be noted that exome studies are still under development in our hospital and therefore the results of this article are based on the results of the panel.

The inclusion criteria were as follows: (1) bilateral congenital cataracts, (2) absence of systemic disease or syndrome potentially related to congenital cataracts. The exclusion criteria were as follows: (1) juvenile development of cataract, (2) unilateral cataract, (3) systemic disease possibly associated with congenital cataracts. Demographic data including date of birth, gender, ethnicity, geographical origin of parents and grandparents, proband past medical history and detailed family history of ocular diseases were collected. Full medical records including all ophthalmological evaluations since diagnosis were reviewed and the following data was collected: type of cataract (anterior polar, anterior subcapsular, lamellar, nuclear, posterior subcapsular, posterior polar, pulverulent) anterior chamber developmental abnormalities (microphthalmia, microcornea, aniridia), vitreous or retinal abnormalities. A sample of 5 mL of peripheral blood was obtained from all probands and, whenever possible from both biological parents, and genomic DNA was extracted from leukocytes.

Genomic DNA was isolated from peripheral blood samples in the preanalytical area in our institute with commercial Chemagic MSM I (Chemagen, PerkinElmer, EEUU). DNA quantity was assessed by spectrofluorometer quantification using TECAN M200 Infinite Pro Microplate Reader. Libraries were prepared using 0.5–1 μg of genomic DNA according to the SeqCap EZ system.

Mutation screening strategy was based on the use of next-generation sequencing (NGS) implementing a panel (OFTv2.1) including 39 known non-syndromic congenital cataracts disease genes as follows: *AGK*, *BFSP1*, *BFSP2*, *CHMP4B*, *CRYAA*, *CRYAB*, *CRYBA1*, *CRYBA4*, *CRYBB1*, *CRYBB2*, *CRYBB3*, *CRYGA*, *CRYGC*, *CRYGD*, *CRYGS*, *EPHA2*, *EYA1*, *FOXE3*, *FTL*, *FYCO1*, *GALK1*, *GCNT2*, *GJA3*, *GJA8*, *HSF4*, *LEPREL1*, *LIM2*, *MAF*, *MIP*, *MIR184*, *NHS*, *PAX6*, *PITX3*, *PXDN*, *SIL1*, *SLC16A12*, *SIX6*, *TDRD7*, *VIM*. The genes were selected based on https://cat-map.wustl.edu (accessed on 20 January 2021, latest update August 2020), an online chromosome map and reference database for inherited cataracts.

OFTv2.1 panel was designed with NimbleDesign software (https://design.nimblegen.com (accessed on 10 January 2020), Roche NimbleGen, Inc., Pleasanton, CA, USA): HG19 NCBI Build 37.1/GRCh37, target bases covered 99.6% and size 370.730 Kb. For each sample, paired-end libraries were created according to the standard NGS protocols KAPA HTP Library Preparation Kit for Illumina^®^ platforms, SeqCap EZ Library SR (Roche NimbleGen, Inc. USA) and NEXTflex-96 Pre Capture Combo Kit (Bioo Scientific, Austin, TX, USA) for indexing. Captured sample DNA was sequenced on a NextSeq 500 instrument (Illumina, Inc., San Diego, CA, USA) using a high cartridge v2, according to the standard operating protocol.

For the analysis of the results, the INGEMM Clinical Bioinformatics team has designed a bioinformatics analysis system leading to the identification of point polymorphisms (SNP), insertions and deletions of small DNA fragments as well as structural variants of larger size in the capture regions included in the next-generation sequencing panels. The system comprises a sample pre-processing step, read alignment against a reference genome, identification and functional annotation of variants and variant filtering. In all these steps, open tools widely used in the scientific community as well as in-house tools are employed. Likewise, all phases are designed in a robust manner including statistic parameters reporting the process status and the convenience of continuing with the analysis. This allows monitoring the process and appropriate quality controls to release a reliable report of the afore-mentioned variants. Finally, the system executes security backups of the raw and processed data. These data are saved in a database with encrypted and anonymized registries to preserve the confidentiality of the patients.

Bioinformatic analysis was carried out by the Clinical Bioinformatics Unit of INGEMM centre. Software:trimmomatic-0.32, bowtie2-align version 2.1.0, picard-tools 1.106, samtools Version.0.1.19-44428cd, bedtools v2.18.1, GenomeAnalysisTK version 3.3-0. Data Bases:dbNSFP version 3.0, dbSNP v138, ClinVar date 20140703, SnpE4.1l, Exac r0.3, SIFT ensembl 66, Polyphen-2 v2.2.2, MutationAssessor, release 2, FATHMM, v2.3, CADD, v1.3 and dbscSNV1.1.

Variants were analyzed for possible pathogenic clinical significance according to the 2015 American College of Medical Genetics and Genomics (ACMG) guidelines [26], based on a combination of previous reports in the literature and computational, functional, and population data. Variants that were classified as pathogenic or likely pathogenic according to the ACMG guidelines were validated using Sanger sequencing in the proband and segregation was performed in family members when possible.

## 6. Conclusions

Although congenital cataracts have a low prevalence, they are a frequent cause of vision loss in infancy because of limited surgical outcomes, other associated ocular abnormalities and the high risk of amblyopia. The causative mutation in non-syndromic congenital cataracts can often be diagnosed with current genetical analysis techniques. In the Spanish population, as in most case series reported so far, mutations in crystallin genes are the leading cause of congenital cataracts. The NGS approach for detecting mutations in congenital cataracts represents an advance in the diagnosis, allowing a better assessment of the risk in a subject´s family and improving future reproductive counselling in patients affected. NGS versus whole exome sequencing might be a better option in most tertiary hospitals of public health systems.

## Figures and Tables

**Figure 1 genes-12-00580-f001:**
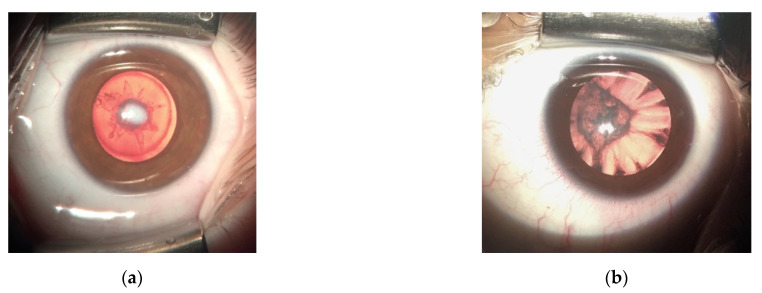
(**a**) Polar anterior congenital cataract; (**b**) Nuclear congenital cataract.

**Figure 2 genes-12-00580-f002:**
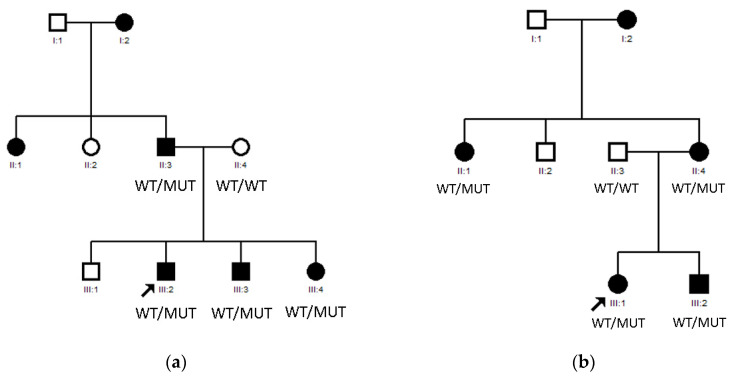
(**a**) Pedigree of Family 17; (**b**) Pedigree of Family 18.

**Figure 3 genes-12-00580-f003:**
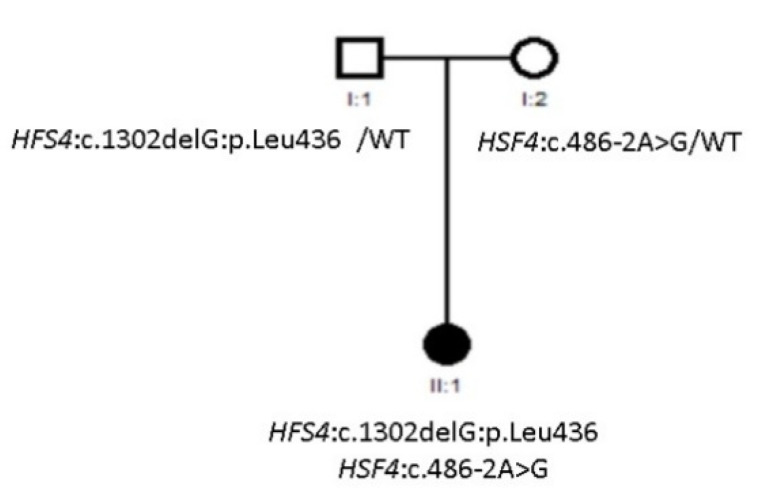
Pedigree of Family 27.

**Figure 4 genes-12-00580-f004:**
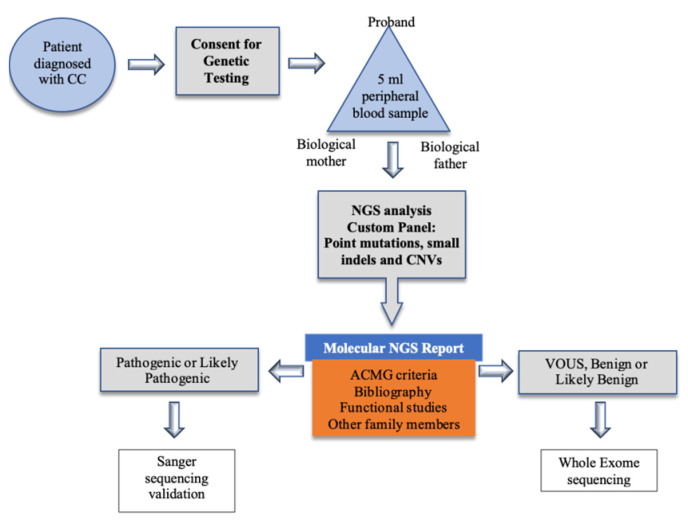
Congenital cataract diagnosis pathway. CC (congenital cataract). NGS (next-generation sequencing).

**Table 1 genes-12-00580-t001:** Congenital cataracts phenotype characteristics.

Family ID	Type of CC	Microphthalmia	Microcornea	Iris Malformations	Family History of CC	Gene
Family 01	Lamellar	-	-	-	-	*CRYBB2*
Family 02	Nuclear	Yes	-	-	-	*CRYBA4*
Family 03	Lamellar	-	-	-	Yes	*CRYGS*
Family 04	Nuclear	Yes	-	-	Yes	*CRYAA*
Family 05	Nuclear	-	-	-	Yes	*CRYGD*
Family 06	Unknown	-	-	-	-	*CRYGD*
Family 07	Nuclear	Yes	-	Yes	-	*CRYGC*
Family 08	Nuclear	-	-	Yes	Yes	*CRYGC*
Family 09	Lamellar	-	-	-	Yes	*CRYBB3*
Family 10	Lamellar	-	-	-	Yes	*GJA3*
Family 11	Nuclear	-	-	-	Yes	*GJA3*
Family 12	Nuclear	-	Yes	-	-	*GJA8*
Family 13	Lamellar	-	-	-	Yes	*GJA8*
Family 14	Nuclear	-	-	-	-	*GJA8*
Family 15	Posterior subcapsular	-	Yes	-	-	*GJA8*
Family 16	Nuclear	Yes	-	-	-	*GJA8*
Family 17	Nuclear	-	-	-	Yes	*LIM2*
Family 18	Lamellar	-	-	-	Yes	*LIM2*
Family 19	Unknown	-	-	-	-	*LIM2*
Family 20	Nuclear	Yes	-	-	Yes	*EPHA2*
Family 21	Unknown	-	-	-	-	*EPHA2*
Family 22	Nuclear	-	-	-	-	*PAX6*
Family 23	Unknown	Yes	Yes	-	-	*PAX6*
Family 24	Nuclear	-	-	-	Yes	*MIP*
Family 25	Nuclear	-	-	-	Yes	*MIP*
Family 26	Nuclear	-	-	-	Yes	*MIP*
Family 27	Nuclear	-	-	-	Yes	*HSF4*
Family 28	Posterior subcapsular	-	-	-	Yes	*PITX3*
Family 29	Lamellar	-	-	-	-	*ABCB6*
Family 30	Posterior polar	-	-	-	-	*TDRD7*

**Table 2 genes-12-00580-t002:** Congenital cataracts NGS (next-generation sequencing) results. Het (heterozygosity). ACMG (American College of Medical Genetics and Genomics); VUS: uncertain significance variant, LP: likely pathogenic and P: pathogenic.

Family ID	Gene	Transcript	Mutation	ACMG Criteria	Variant Type	Zygosity	Segregation Analysis Performed	De Novo/Inherited	Described by
Family 01	*CRYBB2*	NM_000496.2	c.562C>A:p.Arg188Ser	LP	Missense	Het	Yes	De novo	Wang Z et al., 2020
Family 02	*CRYBA4*	NM_001886.3	c.206T>C:p.Leu69Pro	P	Missense	Het	Yes	De novo	Billingsley G et al., 2006
Family 03	*CRYGS*	NM_017541	c.53G>A:p.Gly18Asp	P	Missense	Het	Yes	Maternal	Zhai Y et al., 2017
Family 04	*CRYAA*	NM_000394.4	c.61C>T:p.Arg21Trp	P	Missense	Het	Yes	Paternal	Hansen L et al., 2007
Family 05	*CRYGD*	NM_006891.3	c.T232C:p.Ser78Pro	LP	Missense	Het	Yes	Maternal	Yang G et al., 2016
Family 06	*CRYGD*	NM_006891.3	c.70C>T:p.Pro24Ser	P	Missense	Het	No	Unknown	Plotnikova OV et al., 2007
Family 07	*CRYGC*	NM_020989.4	c.425_432dup:p.Leu145Glyfs * 5	LP	Frameshift	Het	Yes	De novo	Graw J et al., 2002
Family 08	*CRYGC*	NM_020989.4	c.438delG:p.Arg147Glyfs * 32	LP	Frameshift	Het	Yes	Maternal	Novel
Family 09	*CRYBB3*	NM_004076.5	c.531G>T:p.Glu177Asp	VUS	Missense	Het	Yes	Paternal	VCV000900831.1. Variation ID:900831
Family 10	*GJA3*	NM_021954.4	c.595G>A:p.Glu199Lys	LP	Missense	Het	Yes	Maternal	Novel
Family 11	*GJA3*	NM_021954.4	c.817_818insATG:p.Tyr272_Ala273insAsp	LP	In-frame deletion	Het	Yes	Paternal	Novel
Family 12	*GJA8*	NM_005267.5	c.226C>G:p.Arg76Gly	LP	Missense	Het	Yes	De novo	Reis LM et al., 2013
Family 13	*GJA8*	NM_005267.5	c.64G>A:p.Gly22Ser	LP	Missense	Het	Yes	Maternal	Ye Y et al., 2019
Family 14	*GJA8*	NM_005267.5	c.565C>G:p.Pro189Ala	LP	Missense	Het	No	Unknown	Novel
Family 15	*GJA8*	NM_005267.5	c.226C>T:p.Arg76Cys	LP	Missense	Het	Yes	De novo	Reis LM et al., 2013
Family 16	*GJA8*	NM_005267.5	c.592C>T:p.Arg198Trp	P	Missense	Het	Yes	De novo	Hu S et al., 2010
Family 17	*LIM2*	NM_030657.4	c.388C>T:p.Arg130Cys	LP	Missense	Het	Yes	Paternal	Berry V et al., 2020
Family 18	*LIM2*	NM_030657.4	c.388C>T:p.Arg130Cys	LP	Missense	Het	Yes	Paternal	Berry V et al., 2020
Family 19	*LIM2*	NM_030657.4	c.385C>T:p.Arg129Cys	VUS	Missense	Het	Yes	Maternal	Novel
Family 20	*EPHA2*	NM_004431.4	c.2826-9G>A	LP	Splice	Het	Yes	Maternal	Zhang T et al., 2009
Family 21	*EPHA2*	NM_004431.4	c.649G>C:p.Gly217Arg	VUS	Missense	Het	Yes	Paternal	Novel
Family 22	*PAX6*	NM_001258462.3	c.77G>A:p.Arg26Gln	P	Missense	Het	Yes	De novo	Williamson KA et al., 2020
Family 23	*PAX6*	NM_001258462.3	c.219G>T:p.Arg73Ser	LP	Missense	Het	No	Unknown	Novel
Family 24	*MIP*	NM_012064.3	c.676dupC:p.Arg226fs	P	Frameshift	Het	Yes	Paternal	Novel
Family 25	*MIP*	NM_012064.3	c.430T>C:p.Cys144Arg	LP	Missense	Het	Yes	Maternal	Sun W et al., 2020
Family 26	*MIP*	NM_012064.3	c.607-1G>T	LP	Splice	Het	Yes	*De novo*	Sun W et al., 2020
Family 27	*HSF4*	NM_001040667.2	Allele 1: c.486-2A>G Allele 2: c.1302delG:p.Leu436	LP	Allele 1: Splice Allele 2: Frameshift	Compound het	Yes	Allele 1: Maternal Allele 2: Paternal	Novel
Family 28	*PITX3*	NM_005029.3	c.640_656delGCCCTGCAGGGCCTGGG:p.Ala214Argfs * 42	LP	Frameshift	Het	Yes	Paternal	Anand D et al., 2018
Family 29	*ABCB6*	NM_005689.4	c.1762G>A:p.Gly588Ser	VUS	Missense	Het	Yes	Maternal	Saison C et al., 2013
Family 30	*TDRD7*	NM_014290.2	Allele 1: c.1085C>T:p.Pro362Leu Allele 2: Not found	VUS	Allele 1: Missense Allele 2: Not found	Unknown	Yes	Allele 1: Maternal Allele 2: Not found	Novel

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
