# Peer review of "Molecular and Genetic Mechanism of Non-Syndromic Congenital Cataracts. Mutation Screening in Spanish Families"

_genes, 2021, doi:10.3390/genes12040580_

Round 1

Reviewer 1 Report

The authors presented a very nice study of congenital cataract in the Spanish population. They have not only revealed known pathogenic variants but also found novel variants affecting genes associated with congenital cataract. This study significantly expanded the mutation spectrum. 

Nevertheless, I have some comments which should be addressed to the authors. 

  1. Authors noted that segregation analysis was not always performed. However, it is not clear in which cases it was performed and where it was not. I think the Table 1 should be updated with the results of segregation analysis. "De novo"/inherited status of the variants along with the information about sporadic or familial case should also be included at least in the Table 1.
  2. Description of Family 27 with biallelic variants in HSF4 gene is quite confusing. Authors mentioned that "Family number 27 includes 3 affected individuals, belonging to the same generation, as shown in Figure 3." Nevertheless, Figure 3 represents only nuclear family with one affected proband. Further, authors stated thaht "NGS was performed, and a HSF4 (Allele 1: 486-2A>G, Allele 2: 1302delG) homozygous mutation was found in all patients of the presented family." How compound heterozygous mutations could be homozygous? Table 1 has also supported that the proband in Family 27 is compound heterozygous. This part should be rewritten.
  3. Family 30 lacks the second allele in TDRD7 gene. Nevertheless, Table 1 contains information that proband is compound heterozygous. How could it be understood? 
  4. The final statement within the Discussion section refers to the arguments concerning false negative and false positive results. Authors stated that "the false-negative and false-positive rates for the method they used are lower than those found in WES and WGS studies". However, this statement has no support either within the Results or Discussion sections nor by literature citations. It is not clear how authors have eatimated FN and FP rates and made this conclusion. 

There are some minor comments also. English language should be polished throughout the manuscript. List of the references should also be formatted according to the journal style.

Author Response

The authors presented a very nice study of congenital cataract in the Spanish population. They have not only revealed known pathogenic variants but also found novel variants affecting genes associated with congenital cataract. This study significantly expanded the mutation spectrum. 

Nevertheless, I have some comments which should be addressed to the authors. 

  1. Authors noted that segregation analysis was not always performed. However, it is not clear in which cases it was performed and where it was not. I think the Table 1 should be updated with the results of segregation analysis. "De novo"/inherited status of the variants along with the information about sporadic or familial case should also be included at least in the Table 1.

Reply 1: We agree with Reviewer 1, and have modified our table to include the mentioned information about segregation and inheritance.

Changes in the text:We have modified our text as advised, and added the requested columns as “Segregation analysis performed” and “De novo/Inherited” (see Table.1b).

  1. Description of Family 27 with biallelic variants in HSF4 gene is quite confusing. Authors mentioned that "Family number 27 includes 3 affected individuals, belonging to the same generation, as shown in Figure 3." Nevertheless, Figure 3 represents only nuclear family with one affected proband. Further, authors stated that "NGS was performed, and a HSF4 (Allele 1: 486-2A>G, Allele 2: 1302delG) homozygous mutation was found in all patients of the presented family." How compound heterozygous mutations could be homozygous? Table 1 has also supported that the proband in Family 27 is compound heterozygous. This part should be rewritten.

Reply 2: As reviewer 1 points out, we did indeed make a mistake. The members of family 27 who have been tested include a nuclear family with one affected proband, nevertheless there are other affected individuals belonging to same generation as the proband who have not been tested yet. Regarding the inheritance pattern and the type of mutation it is compound heterozygous.

Changes in the text: We have modified our text as advised, and completely rewritten information about Family 27 (see Page 6-7, lines 448-481).

  1. Family 30 lacks the second allele in TDRD7 gene. Nevertheless, Table 1 contains information that proband is compound heterozygous. How could it be understood?

Reply 3: Reviewer 1 is right and the information written about “Family 30” in Table 1 was inaccurate. This information has already been changed in the text.

Changes in the text: We have modified our text as advised (see Family 30, Table.1b).

  1. The final statement within the Discussion section refers to the arguments concerning false negative and false positive results. Authors stated that "the false-negative and false-positive rates for the method they used are lower than those found in WES and WGS studies". However, this statement has no support either within the Results or Discussion sections nor by literature citations. It is not clear how authors have estimated FN and FP rates and made this conclusion. 

Reply 4: We agree with Reviewer 1, as we have drawn this conclusion from the use of other panels at NGS, but it is a fact that we lack the data to achieve this conclusion.

Changes in the text: We have modified our text as advised, and deleted paragraph mentioned (see Page 9, lines 634-637).

There are some minor comments also. English language should be polished throughout the manuscript. List of the references should also be formatted according to the journal style.

Reply: The English language version of the publication has been checked again with an expert. Thank you very much for the comments and the review, it has helped us to improve the article and the research.

Reviewer 2 Report

This paper has the important purpose “to identify mutations responsible for non-syndromic congenital cataracts (CC) through the implementation of next-generation sequencing (NGS) in our centre”. However, I am not sure that this justifies publication of the paper nor that the current manuscript adds any useful insight to the molecular or genetic mechanisms of non-syndromic congenital cataracts. There are few new mutations identified (and they are in known cataract genes. There is no insight given regarding the variants of unknown significance affecting ABCB6, RECQL4, TDRD7 and collagen genes (COL9A2, 26 COL8A2, COL11A1). What about the large number of families and individuals where no responsible mutations were identified? The paper would be more interesting if these cases were further studied leading to new information.

Additional specific comments:

  1. Essentially all of the data presented are in Table 1. Yet, this table is turned sideways and is in unacceptably small print. The authors must fix these issues. Perhaps they and reorganize it, break it up into parts, delete some unnecessary columns, etc.
  2. There are unnecessary abbreviations like CC.
  3. The numbers and statistics are presented in confusing and inconsistent ways. For example, the abstract states “Of the probands with an identified variant, 23.7% had mutations in crystallin genes…” Does this mean that they found .237 X .49 X 62 = 6.5 crystallin mutations? 0r, .237 X .49 X 51 = 5.9 crystallin mutations. Either way, this is less than the membrane proteins. The membrane proteins should include LIM2 and MIP as well as the GJAs.
  4. What do the authors mean by the novo? de novo??
  5. What evidence supports the conclusion that “NGS seems to be a valid option for genetic evaluation in patients with CC”?
  6. The Introduction includes a lot of extraneous information regarding the lens etc. It should be revised and focused.

Author Response

Review 2

This paper has the important purpose “to identify mutations responsible for non-syndromic congenital cataracts (CC) through the implementation of next-generation sequencing (NGS) in our center”. However, I am not sure that this justifies publication of the paper nor that the current manuscript adds any useful insight to the molecular or genetic mechanisms of non-syndromic congenital cataracts. There are few new mutations identified (and they are in known cataract genes. There is no insight given regarding the variants of unknown significance affecting ABCB6, RECQL4, TDRD7 and collagen genes (COL9A2, 26 COL8A2, COL11A1). What about the large number of families and individuals where no responsible mutations were identified? The paper would be more interesting if these cases were further studied leading to new information.

Reply:First of all, thank you very much for your comment. We are currently carrying out the study of families with a negative result by exome and genome. At the moment there are no published studies on the congenital cataract spectrum in our country, so we thought it would be interesting to provide data on the Spanish population.

Additional specific comments:

  1. Essentially all of the data presented are in Table 1. Yet, this table is turned sideways and is in unacceptably small print. The authors must fix these issues. Perhaps they can reorganize it, break it up into parts, delete some unnecessary columns, etc

Reply 1: Authors agree with Reviewer 2, and think that the table may be too large and difficult to read.

Changes in the text: We have modified our text as advised, and divided Table 1 into two tables: Table 1.a and Table 1.b (see Page 6).

  1. There are unnecessary abbreviations like CC.

Reply 2: By using the abbreviation of CC our aim was to facilitate the reading.

Changes in the text: We have modified our text as advised and eliminated the abbreviation “CC”.

  1. The numbers and statistics are presented in confusing and inconsistent ways. For example, the abstract states “Of the probands with an identified variant, 23.7% had mutations in crystallin genes…” Does this mean that they found .237 X .49 X 62 = 6.5 crystallin mutations? 0r, .237 X .49 X 51 = 5.9 crystallin mutations. Either way, this is less than the membrane proteins. The membrane proteins should include LIM2 and MIP as well as the GJAs.

Reply 3: Authors have rewritten and recalculated figures to make sure there is no mistake.

Changes in the text: We have modified our text as advised, and rewritten the sentence so its meaning remains clear (see Page 3-4, line 108-268).

  1. What do the authors mean by the novo? de novo??

Reply 4: Reviewer 2 is right and the spelling mistake has been corrected. 

Changes in the text: We have modified our text as advised, and replaced all “the novo” with “de novo”.  

  1. What evidence supports the conclusion that “NGS seems to be a valid option for genetic evaluation in patients with CC”?

Reply 5: Several papers published in the past 10 years have proven that targeted NGS in presumed nonsyndromic congenital cataract patients provides significant diagnostic information in both familial and sporadic cases. Some examples are:

Ma AS, Grigg JR, Ho G, Prokudin I, Farnsworth E, Holman K, Cheng A, Billson FA, Martin F, Fraser C, Mowat D, Smith J, Christodoulou J, Flaherty M, Bennetts B, Jamieson RV. Sporadic and Familial Congenital Cataracts: Mutational Spectrum and New Diagnoses Using Next-Generation Sequencing. Hum Mutat. 2016 Apr;37(4):371-84. doi: 10.1002/humu.22948. Epub 2016 Jan 14. PMID: 26694549; PMCID: PMC4787201.

Astiazarán MC, García-Montaño LA, Sánchez-Moreno F, Matiz-Moreno H, Zenteno JC. Next generation sequencing-based molecular diagnosis in familial congenital cataract expands the mutational spectrum in known congenital cataract genes. Am J Med Genet A. 2018 Dec;176(12):2637-2645. doi: 10.1002/ajmg.a.40524. Epub 2018 Nov 18. PMID: 30450742.

Changes in the text: No changes have been made.

  1. The Introduction includes a lot of extraneous information regarding the lens etc. It should be revised and focused.

Reply 6: Authors agree with Reviewer 2, introduction might be too long and should be shortened.

Changes in the text: We have modified our text as advised, and shortened the introduction (see Page 2-3, lines 57-97).

Authors provide an important study of molecular-genetic causes of congenital cataracts. The detection rate was 49%, that shows that other mechanisms are involved in formation of cataracts.

Very interesting is conclusion that de novo occurrence was recorded in 16 families.

Reply: Thank you very much for the comments and the review, it has helped us to improve the article and the research.

Reviewer 3 Report

Authors provide an important study of molecular-genetic causes of congenital cataracts. The detection rate was 49%, that shows that other mechanisms are involved in formation of cataracts.

Very interesting is conclusion that de novo occurrence was recorded in 16 families.

Major comments

Introduction is too long. Authors should decide between review and article. As this paper was submitted as article, please shorten to 1-2 pages, some information can be moved into discussion.

Figure 1 is missing, probably Figure 2 has wrong label.

Please mark families with de novo occurrence in Table 1 – for this, please confirm de novo occurrence with paternal and maternal analysis.

Table 1 – descriptions on protein level of all frameshifting mutations are wrong. Please follow HGVS standards http://varnomen.hgvs.org/ (for example, mutation in Family 7 - p.(Leu145Glyfs*5) instead of p.Leu145fs).

Very interesting is autosomal dominant inheritance of LIM2 mutations, that has been described recently (Berry et al, 2020). Please provide more information about family 19 as this mutation seems to be novel, with AD inheritance.   

Pathogenic or likely pathogenic mutation were detected in 25 families and in 7 families VOUS was found, but in table 1 is 30 families with detected mutation.  

Mutations in collagen genes are mentioned in abstract and results section (line 207-208) but not in Table 1. Please provide description of this mutations.

I suggest to change “NP” in table 1 into “novel” as these mutations are novel (not published before) and to add this information into abstract. Additionally, I suggest to use P for pathogenic, LP for likely pathogenic and VOUS instead in 3-4-5 classification.

Do not see any column with type of inheritance (AD, AR, ?), so this can be omitted from description.

Figure 2 – please mark who has been tested by adding WT/MUT or WT/WT into pedigrees.

Line 241 - Family number 27 includes 3 affected individuals, belonging to the same generation, as shown in Figure 3. According to Figure 3, there is only one affected individual, please check.

Minor comments

Extra space in first paragraph of Introduction

Line 186 missing space before 16

Author Response

Review 3

Major comments

Introduction is too long. Authors should decide between review and article. As this paper was submitted as article, please shorten to 1-2 pages, some information can be moved into discussion.

Reply: Authors agree with Reviewer 3, the introduction might be too long and should be shortened.

Changes in the text: We have modified our text as advised, and shortened the introduction (see Page 2-3, lines 57-97) and enriched the discussion (see Page 7-9, pages 507-556; pages 595-609).

Figure 1 is missing, probably Figure 2 has wrong label.

Reply: Figure 1 was mislabeled as Figure 2.

Changes in the text: We have modified the name of Figure 2 to Figure 1 (see Page 3, line 83).

Please mark families with de novo occurrence in Table 1 – for this, please confirm de novo occurrence with paternal and maternal analysis.

Reply: We agree with Reviewer 3, and think that our table should be modified to include information about segregation and inheritance.

Changes in the text:We have modified our text as advised, and added the requested columns as “Segregation analysis performed” and “De novo/Inherited” (see Table.1b).

Table 1 – descriptions on protein level of all frameshifting mutations are wrong. Please follow HGVS standards http://varnomen.hgvs.org/ (for example, mutation in Family 7 - p.(Leu145Glyfs*5) instead of p.Leu145fs).

Reply: Authors agree with Reviewer 3, HGVS standards should be followed.

Changes in the text: We have modified our text as advised (Table 1.b).

Very interesting is autosomal dominant inheritance of LIM2 mutations, that has been described recently (Berry et al, 2020). Please provide more information about family 19 as this mutation seems to be novel, with AD inheritance.

Reply: We have modified the table to add these data, as well as the bibliography.

Changes in the text: We have modified our text as advised (Table 1.a and 1.b).

Pathogenic or likely pathogenic mutations were detected in 25 families and in 7 families VOUS was found, but in table 1 is 30 families with detected mutation.  

Reply: Authors decided to mention some of the VOUS variants in the table, but we do agree that it can be confusing. Therefore we decided to delete that data from the table.

Changes in the text: We have modified our text as advised (Table 1.a and 1.b).

Mutations in collagen genes are mentioned in abstract and results section (line 207-208) but not in Table 1. Please provide description of these mutations.

Reply: Authors apologize to Reviewer 3, mutations in collagen were results of the first investigation attempt and were transferred to the manuscript by mistake.

Changes in the text: We have modified our text as advised, and eliminated the wrong content (see Page 5, lines 327). 

I suggest to change “NP” in table 1 into “novel” as these mutations are novel (not published before) and to add this information into abstract. Additionally, I suggest to use P for pathogenic, LP for likely pathogenic and VOUS instead in 3-4-5 classification.

Reply: We agree with Reviewer 3, and think that our table should be modified to include this information.

Changes in the text: We have modified our text as advised (see Table.1b).

Do not see any column with type of inheritance (AD, AR, ?), so this can be omitted from description.

Reply: The column was deleted due to lack of space in Table 1.

Changes in the text: We have modified our text as advised (see Page 7, lines 118-119).

Figure 2 – please mark who has been tested by adding WT/MUT or WT/WT into pedigrees.

Reply: We agree with Reviewer 3 and pedigrees should be modified.  

Changes in the text: We have modified our text as advised (see Page 8, Figures 2 and 3).

Line 241 - Family number 27 includes 3 affected individuals, belonging to the same generation, as shown in Figure 3. According to Figure 3, there is only one affected individual, please check.

Reply: As reviewer 3 points out, we did indeed make a mistake. The members of family 27 who have been tested include a nuclear family with one affected proband, nevertheless there are other affected individuals belonging to same generation as the proband who have not been tested yet. Regarding the inheritance pattern and the type of mutation it is compound heterozygous.

Changes in the text: We have modified our text as advised, and completely rewritten information about Family 27 (see Page 6-7, lines 448-481).

Minor comments

Extra space in first paragraph of Introduction

Line 186 missing space before 16

Changes in the text: Minor comments have been taken into consideration and modified in the text.Thank you very much for the comments and the review, it has helped us to improve the article and the research.

Round 2

Reviewer 1 Report

Dear authors,

Thank you for your responses and the revision of the manuscript. It became much clearer after your corrections.

Some English polishing is still needed for the manuscript, e.g. "no-sense mediated decay" -> "nonsense-mediated decay"; "heterozigous mutation" -> "heterozygous mutation"; "all of them Caucasian Europeans" -> "all of them are Caucasian Europeans"; "PAX-6" -> "PAX6"; etc. Please, check the spelling once again.

I have also some additional questions. PAX6 cases are very interesting in the context of non-syndromic congenital cataracts. Even if PAX6 mutations do not lead to iris malformations, they usually should be associated at least with the foveal hypoplasia. Was any ophthalmological examination of the fundus performed in these cases?

I am also a little bit confused about Figure 4. This figure presents the WES step as one of the stages of the congenital cataract diagnosis pathway. Nevertheless, no patients were tested by WES in the current study. Maybe, it will be more correct to highlight and depict that WES is not performed and is the following step of the DNA diagnosis.

Author Response

Thank you very much for the new comments.

Some English polishing is still needed for the manuscript, e.g. "no-sense mediated decay" -> "nonsense-mediated decay"; "heterozigous mutation" -> "heterozygous mutation"; "all of them Caucasian Europeans" -> "all of them are Caucasian Europeans"; "PAX-6" -> "PAX6"; etc. Please, check the spelling once again.

Reply: The authors have further revised the spelling. We have marked in red the text that have been changed in the manuscript.

I have also some additional questions. PAX6 cases are very interesting in the context of non-syndromic congenital cataracts. Even if PAX6 mutations do not lead to iris malformations, they usually should be associated at least with the foveal hypoplasia. Was any ophthalmological examination of the fundus performed in these cases?

Reply: Although there is an association between aniridia and PAX6 with foveal hypoplasia; al patients included in the present study wete found to have a normal fundus exam. The authors are preparing another manuscript on the different phenotypes associated with PAX6.

I am also a little bit confused about Figure 4. This figure presents the WES step as one of the stages of the congenital cataract diagnosis pathway. Nevertheless, no patients were tested by WES in the current study. Maybe, it will be more correct to highlight and depict that WES is not performed and is the following step of the DNA diagnosis.

Reply: Admittedly confusing, the authors have changed the text explaining figure 4 by highlighting the fact that this publication talks about panels and that the future step will be exoma.

Reviewer 2 Report

The authors have made many of my requested changes

Author Response

Thank you very much!

Reviewer 3 Report

Authors have significantly improved the manuscript.

I have spotted a few typing errors:

Line 94 "...CRYGD andCRYGC),. " - please add space and delete comma

Line 159 heterozigous mutation - heterozygous

Line 122 - VUS or VOUS: uncertain significance variant - only VUS is in table 1, please check.

Author Response

Thank you very much for the new comments.

I have spotted a few typing errors:

Line 94 "...CRYGD andCRYGC),. " - please add space and delete comma

Line 159 heterozigous mutation – heterozygous

Line 122 - VUS or VOUS: uncertain significance variant - only VUS is in table 1, please check.

Reply: Thank you very much. the authors have corrected these errors.

Changes in the text: Line 94, line 159 and line 122, are marked in red.